# Effects of Long-Term Dietary Protein Restriction on Intestinal Morphology, Digestive Enzymes, Gut Hormones, and Colonic Microbiota in Pigs

**DOI:** 10.3390/ani9040180

**Published:** 2019-04-20

**Authors:** Defu Yu, Weiyun Zhu, Suqin Hang

**Affiliations:** Laboratory of Gastrointestinal Microbiology, Nanjing Agricultural University, Nanjing 210095, China; yudefu1225@outlook.com (D.Y.); zhuweiyun@njau.edu.cn (W.Z.)

**Keywords:** pigs, low-protein diet, digestive enzymes, intestinal morphology, gut hormone, colonic microbiota

## Abstract

**Simple Summary:**

In China, a shortage of protein resources is an important limiting factor to the economic benefit of pig production, and the use of protein-restriction diets balanced with amino acids is an effective strategy to save protein resources. However, long-term protein-restriction diets can impair the growth performance, and the reason is still unknown. This study is to investigate the response of gastrointestinal physiology and gut microbiota to the condition of long-term low-protein diet and to try to provide a theoretical foundation for better use of protein resources in swine production. Results showed that presented with moderate protein-restriction diets, pigs are able to adjust their absorption and consumption of nutrients to maintain growth performance; whereas extremely low-protein diets suppress pigs’ appetite, impair intestinal morphology, decrease *Lactobacillus* and *Streptococcus*, and reduce energy expenditure. Thus, moderate reduction of dietary protein is more suitable for pig production than extremely low-protein diets supplemented with essential amino acids, and moderate protein-restriction diets can potentially increase protein utilization in pig production.

**Abstract:**

Using protein-restriction diets becomes a potential strategy to save the dietary protein resources. However, the mechanism of low-protein diets influencing pigs’ growth performance is still controversial. This study aimed to investigate the effect of protein-restriction diets on gastrointestinal physiology and gut microbiota in pigs. Eighteen weaned piglets were randomly allocated to three groups with different dietary protein levels. After a 16-week trial, the results showed that feeding a low-protein diet to pigs impaired the epithelial morphology of duodenum and jejunum (*p* < 0.05) and reduced the concentration of many plasma hormones (*p* < 0.05), such as ghrelin, somatostatin, glucose-dependent insulin-tropic polypeptide, leptin, and gastrin. The relative abundance of *Streptococcus* and *Lactobacillus* in colon and microbiota metabolites was also decreased by extreme protein-restriction diets (*p* < 0.05). These findings suggested that long-term ingestion of a protein-restricted diet could impair intestinal morphology, suppress gut hormone secretion, and change the microbial community and fermentation metabolites in pigs, while the moderately low-protein diet had a minimal effect on gut function and did not impair growth performance.

## 1. Introduction

In China, a shortage of protein resources is an important limiting factor to the economic benefit of pig production [1]. In 2016, China’s soybean imports were 8391 million tons and accounted for more than 26% of the worldwide production, while high-protein (HP) diets led to excretion of excess nitrogen in feces and urine, resulting in low efficiency of nitrogen utilization and environmental pollution. The National Research Council (NRC, 1998) recommends that the requirement of crude protein is 20%, 18%, and 16% for weaned piglets, growing pigs, and finishing pigs, respectively. Previous researches showed that reducing the dietary protein level by less than 4% based on the NRC (1998), supplemented with Lys, Met, Thr, and Trp, did not reduce growth performance of weanling, growing, and finishing pigs [2,3,4,5]. Thus, reducing dietary protein level is an effective strategy to save the protein resource and decrease the emission of nitrogen in urea and feces without impairing the growth performance in pigs. However, the requirement of crude protein recommended by the latest edition of NRC (2012) is 2–4% lower than that of NRC (1998). Whether the dietary protein level can be further reduced based on NRC (2012) and the responses of growth performance, gut development, and microbiota to low-protein diets are unclear.

The intestinal epithelium morphology is the structural basis for digestive and absorptive functions. Villous height in weaned piglets was not affected by the reduction of dietary crude protein from 21% to 17% when supplemented with Lys, Met, Thr, Trp, Ile, and Val [6]. Another research supplemented eight essential amino acids in a low-protein diet with crude protein (CP) levels reduced from 23.1% to 18.9% and did not note villous atrophy in piglets after two weeks [7]; however, further reduction to 17.2% was associated with impaired villous height in both the duodenum and jejunum. Whether long-term protein restriction impairs the villous morphology of pigs remains unknown.

Before dietary protein is absorbed by the villous, it is broken down into peptides and amino acids by proteinase in the gastrointestinal tract (GIT). The production of pepsin from pepsinogen is induced by gastric hydrochloric acid (HCl) regulated by the enteric endocrinal systems [8]. Somatostatin (SS), which secretes from the D cells of the gastric oxyntic and pyloric mucosa, directly inhibits HCl secretion from parietal cells and indirectly inhibits gastrin secretion from G cells. Gastrin acts as the principal secretagogue of HCl and stimulates both acid secretion and negative feedback inhibition of acid secretion via SS. H^+^-K^+^-ATPase is expressed in the parietal cells, where it mediates the electro-neutral exchange of intracellular H^+^ and extracellular K^+^ to achieve acid secretion. However, information about the interaction between dietary protein level and enteric endocrine hormone is limited, especially for pigs fed a long-term low-protein diet.

The pig GIT harbors trillions of commensal bacteria that play a major role in the health of the host. The community and metabolic activities of gut microbiota are affected by dietary protein level composition [9,10,11]. Weaning pigs fed CP levels reduced by 23% to 17% with Lys, Met, Thr, and Trp supplementation had no effect on microbial populations in the ileal and colonic digesta [12]. However, another study showed that weaning pigs fed CP levels reduced by 20% to 14% with balanced Lys, Met, Thr, and Trp reduced Shannon indices of bacterial diversity and the number of *Firmicutes* and *Clostridium Cluster IV* in the cecal digesta, decreased concentrations of cecal ammonia, and reduced concentrations of acetate and branch chain fatty acids [13]. A reduction of dietary CP levels from 22.5% to 17.6% for weaning pigs decreased colonic ammonia, but had no effects on short chain fatty acids [14]. Considering the various complex metabolites in the hindgut and their potential roles in the host, an understanding of the influence of long-term low-protein diets on gut microbiota and metabolites is needed.

Gut homeostasis is important for pig growth. Thus, the aim of this study was to investigate the effects of long-term dietary protein restriction on intestinal morphology, digestive enzymes, gut hormones, and colonic microbiota in pigs, which may provide a theoretical explanation for the change of growth performance under the condition of reducing dietary protein. This study is expected to provide a good foundation for future diet formula in practical use and contributes to saving a large quantity of protein.

## 2. Materials and Methods

This study was conducted and approved by the Nanjing Agricultural University Animal Care and Use Committee. The license number is SYXK-2017-0027 with a period of validity to 19 June 2022.

### 2.1. Animals, Experimental Design, and Diet

Eighteen Duroc × Landrace × Yorkshire weaned barrows (35 days of age, average body weight of 9.46 ± 0.61 kg) were randomly assigned using a randomized block design into a normal protein (NP) diet group, a moderately low-protein (MP) diet group, in which protein was reduced by 3% compared to that of the NP group, and a low-protein (LP) diet group, in which protein was reduced by 6% compared to the NP group (Table 1). Each group consisted of six replicates, and the trial lasted 16 weeks. Experimental diets in the MP and LP groups were supplemented with four essential amino acids (L-lysine, L-methionine, L-threonine, and L-tryptophan) to meet the requirements of weaned, growing, and finishing pigs according to NRC (2012). Pigs were maintained individually in metabolic cages with free access to feed and drinking water.

### 2.2. Blood Biochemical Parameters and Hormone Analysis

After a 16-week trial, blood samples were collected by jugular venipuncture after fasted for 12 h, followed by 3000× *g* at 4 °C for 15 min; serum was separated and immediately stored at −20 °C for further biochemical parameter analysis. Serum biochemical parameters, including total protein, blood urea nitrogen, glucose, cholesterol, and triglycerides, were evaluated using commercial kits according to the manufacturers’ instructions (Nanjing Angle Gene Biotechnology, Nanjing, China). Blood hormones of serum were analyzed using commercially-available ELISA kits specific to porcine tissues, according to the manufacturers’ instructions (Nanjing Angle Gene Biotechnology, Nanjing, China).

### 2.3. Digestive Enzyme and Intestinal Morphology

Pigs fasted overnight were anesthetized with pentobarbital sodium (50 mg/kg body weight). After jugular exsanguination, the abdomen was incised, and the GIT was immediately removed and rinsed with saline (0.9% NaCl). The digesta of the stomach, jejunum, ileum, and colon were separately collected and stored at −20 °C. The intestinal tissues of the duodenum, jejunum, and ileum (~2 cm long) were obtained and stored in a 10% formaldehyde solution. The villous height and crypt depth were determined. Briefly, each tissue sample was used to prepare five slices of three sections (5 µm thick), which were stained in hematoxylin and eosin with intact and well-oriented crypt-villus units selected for intestinal morphology analysis (Scion Image Software, Version 4.02, 2004, Meyer Instruments, Inc., Houston, TX, USA). The activities of gastric pepsin and trypsin in the digesta were analyzed using commercially-available pepsin and trypsin assay kits according to the manufacturer’s instructions (ANG-SH-21041 and ANG-SH-21052, Nanjing Angle Gene Biotechnology, Nanjing, China).

### 2.4. Determination of Short-Chain Fatty Acids, Ammonia-N, and Biogenic Amines

Colonic luminal contents were collected into separate plastic bags for full mixture. One hundred grams of mixed colonic contents were taken and stored in liquid nitrogen for the analysis of the microbial community. The remaining contents were stored at −20 °C to measure the fermentation metabolites, including SCFAs, ammonia-N (NH_3_-N), and biogenic amines.

Three grams of each sample were weighed into a 10-mL tube and mixed with 1 mL of 25% (*w*/*v*) metaphosphoric acid and 6 mL of water. Samples were then centrifuged at 17,000× *g* for 10 min, and the supernatant was removed to analyze SCFA concentration using a capillary column gas chromatograph (GC-14B, Shimadzu, Japan; capillary column: 30 m × 0.32 mm × 0.25 mm film thickness).

The concentrations of biogenic amines in the digesta were determined using the method described by [15]. Briefly, 200 mg of the colonic content was homogenized with 1 mL trichloroacetic acid to precipitate proteins. The mixed solution was centrifuged at 3600 rpm for 10 min, and the supernatant was transferred to a new tube and mixed with an equal volume of n-hexane for fat extraction. This procedure was repeated several times. Then, 20 μL of internal standard, 1.5 mL of saturated sodium bicarbonate solution, 1 mL of dansyl chloride, and 1 mL of NaOH were added to the pretreated sample and vortexed at 60 °C for 45 min. One hundred milliliters of ammonia were added to the mixture to stop the reaction. Finally, the sample was extracted with diethyl ether, dried, and re-dissolved in acetonitrile for injection. The pretreated samples were analyzed by high performance liquid chromatography (HPLC) (Agilent EE00, Palo Alto, CA, USA) equipped with an ultraviolet (UV) detector (Waters 2998, 254 nm, Agilent Technologies, Palo Alto, CA, USA) and a reversed phase column Zorbax Extend-C18 (Agilent Technologies, Palo Alto, CA, USA), 5 mm (250 × 4.6 mm).

For NH_3_-N determination, one gram of colonic digesta was transferred into a 1.5-mL Eppendorf tube, and 1 mL of 0.2 M HCl was added to acidify the samples. After homogenization, the mixed solution was centrifuged at 4000 rpm for 10 min; then, a 40-µL supernatant was removed and acidified with 0.96 mL of 0.2 M HCl and stored in a freezer (20 °C) for NH_3_-N analysis. NH_3_-N concentration was measured using the indophenol method [16].

### 2.5. Illumina MiSeq Sequencing

Universal primers targeting V3 to V4 variable regions of the bacterial 16S rRNA gene were applied for PCR amplification and subsequent Illumina MiSeq sequencing to analyze the microbial community. The forward primer was 341F 5′-barcode- CCTAYGGGRBGCASCAG-3′, and the reverse primer was 806R 5′-GGACTACNNGGGTATCTAAT-3′ [17], for which the barcode was an eight-based sequence unique to each sample. The PCR cycling protocol consisted of an initial denaturation at 95 °C for 3 min, followed by 27 cycles at 95 °C for 30 s, 55 °C for 30 s, 72 °C for 45 s, and a final extension at 72 °C for 10 min. A 20-μL mixed reaction was composed of 0.8 μL of 5 μM of each primer, 0.4 μL of Pfu polymerase, 2 μL of 2.5 mM dNTPs, 4 μL of 5-fold Pfu buffer (TransGen Biotech, Shanghai, China), and 10 ng of template DNA. The PCR amplicons were separated by 2% agarose gel electrophoresis and purified using the AxyPrep DNA Gel Extraction Kit (Axygen Biosciences, Shanghai, China). The concentration of PCR products was quantified using QuantiFluor™-ST (Promega, Madison, WI, USA). Then, amplicon pyrosequencing on the IlluminaMiSeq platform was carried out as recommended in the manufacturer’s instructions (Shanghai Technology Majorbio Bio-Pharm Co., Ltd., Shanghai, China).

Universal primers targeting V3 to V4 variable regions of the bacterial 16S rRNA gene were applied for PCR amplification and subsequent Illumina MiSeq sequencing to analyze the microbial community. The forward primer was 341F 5′-barcode- CCTAYGGGRBGCASCAG-3′, and the reverse primer was 806R 5′-GGACTACNNGGGTATCTAAT-3′ [17], for which the barcode was an eight-based sequence unique to each sample. The PCR cycling protocol consisted of an initial denaturation at 95 °C for 3 min, followed by 27 cycles at 95 °C for 30 s, 55 °C for 30 s, 72 °C for 45 s, and a final extension at 72 °C for 10 min. A 20-μL mixed reaction was composed of 0.8 μL of 5 μM of each primer, 0.4 μL of Pfu polymerase, 2 μL of 2.5 mM dNTPs, 4 μL of 5-fold Pfu buffer (TransGen Biotech, Shanghai, China), and 10 ng of template DNA. The PCR amplicons were separated by 2% agarose gel electrophoresis and purified using the AxyPrep DNA Gel Extraction Kit (Axygen Biosciences, Shanghai, China). The concentration of PCR products was quantified using QuantiFluor™-ST (Promega, Madison, WI, USA). Then, amplicon pyrosequencing on the IlluminaMiSeq platform was carried out as recommended in the manufacturer’s instructions (Shanghai Technology Majorbio Bio-Pharm Co., Ltd., Shanghai, China).

### 2.6. Sequencing Data Analysis

After sequencing, all reads were filtered using QIIME (Version 1.17, Majorbio, Shanghai, China) for quality control. Reads shorter than 50 bp that contained one or more ambiguous bases and two nucleotide mismatches in primer matching were removed. The reads that overlapped longer than 10 bp were assembled. The operational taxonomic units of the effective sequences were carried out based on a 97% similarity cutoff using UPARSE (Version 7.1, Majorbio, Shanghai, China). The chimeric sequences were identified and removed using UCHIME (Majorbio, Shanghai, China). The phylogenetic affiliation of each 16S rRNA gene sequence was analyzed using the RDP Classifier (http://rdp.cme.msu.edu/) with a standard minimum support threshold of 80% [18]. The coverage was calculated to evaluate the sampling effort using Good’s method [19]. The richness estimator (using ACE and Chao indices) and the diversity estimator (using Shannon and Simpson indices) were analyzed using the Mothur program [20].

### 2.7. Quantification of Bacterial Populations Using Real-Time PCR

The bacterial groups, including total bacteria and the genes of *Lactobacillus*, *Bacteroides*, *Ruminococcus*, *Clostridium Cluster IV*, *Enterobacteriaceae*, and *Clostridium Cluster XIV*, were quantified using real-time PCR (qPCR) with specific primers (Appendix A
Appendix A) and SYBR Green PCR Mastermix (Applied Biosystems, Foster City, CA, USA) in the StepOnePlus real-time PCR system (Life Technologies, Carlsbad, CA, USA), as previously described [21].

### 2.8. Statistical Analysis

Statistical analyses were performed using a one-way analysis of variance (ANOVA) with Statistical Software Package (SPSS) 16.0 (IBM, Armonk, NY, USA). The Student–Newman–Keuls multiple range test was employed to compare differences among treatment means. Differences at *p* < 0.05 were considered significant.

## 3. Results

### 3.1. Growth Performance

From weeks 5 to 16, low-protein (LP) diets significantly reduced body weight, average daily gain and feed intake in piglets, growing, and finishing pigs compared with normal protein (NP) diets. The ratio of feed intake to weight gain was decreased in piglets and growing pigs in low-protein (LP) diets. However, moderately low-protein (MP) diets had no effect on body weight, average daily gain, and feed intake during the whole period in pigs (the detailed data belonging to the Institute of Subtropical Agriculture in China).

### 3.2. Blood Biochemical Indices

Reducing dietary protein levels had no effect on the concentration of glucose in the plasma (*p* > 0.05). Blood urea nitrogen was significantly decreased in pigs fed LP diets compared to those fed NP diets (*p* < 0.05; Table 2). The concentration of cholesterol in the plasma increased as protein decreased in the LP group (*p* < 0.05; Table 2). Additionally, the concentration of total protein, albumin, globulin, and triglyceride in the plasma was not affected by different dietary protein levels (*p* > 0.05; Table 2).

### 3.3. Digestive Enzyme Activity

Stomach pepsin activity increased in the LP group compared to the NP group (*p* < 0.05), while H^+^-K^+^-ATPase enzyme activity in the LP group increased (*p* < 0.05; Table 3). The dietary protein level had no effect on the activity of jejunal trypsin and ileal trypsin (*p* > 0.05; Table 3).

### 3.4. Intestinal Morphology

The LP diets reduced the villous height and the crypt depth in the duodenum compared to those in the normal group (*p* < 0.05; Table 4); however, the ratio of the villous height to crypt depth was not affected in the duodenum. In contrast, there was a reduction of the ratio of the villous height to crypt depth in the jejunum as the dietary protein was decreased in the LP group (*p* < 0.05; Table 4). Reducing the dietary protein increased the crypt depth in the jejunum (*p* = 0.082) and decreased the villous height (*p* = 0.063) and the crypt depth in the ileum (*p* = 0.091; Table 4).

### 3.5. Gut Hormones

Both MP and LP diets significantly decreased serotonin (5-HT), ghrelin, somatostatin (SS), gastrin, and glucose-dependent insulinotropic polypeptide (GIP) concentrations in plasma compared to the NP diet (*p* < 0.05; Table 5). Pigs fed an MP diet had significantly increased concentrations of cholecystokinin (CCK) compared to those fed NP and LP diets (*p* < 0.05; Table 5). Peptide tyrosine tyrosine (PYY) was the same among the three groups. Leptin significantly decreased in the plasma from pigs in the LP group compared to those in the NP and MP groups (*p* < 0.05; Table 5).

### 3.6. Colonic Microbial Counts

The richness and diversity estimator of microbiota were not affected by dietary protein level in the colonic luminal contents of pigs (Appendix A
Appendix A). At the genus level, the MP and LP diets significantly decreased the relative abundance of *Lactobacillus* and *Turicibacter* compared to those in the colon in the LP group (*p* < 0.05; Table 6; Figure 1). The relative abundance of *Streptococcus* was reduced in the LP group compared to the NP group (*p* < 0.05; Table 6; Figure 1). Significantly higher relative abundance of *Prevotella* and *Lachnospira* were observed in the pigs fed the LP and MP diets compared to those fed the NP diet (*p* < 0.05; Table 6; Figure 1). However, the LP diet rather than the MP diet increased the relative abundance of *Dorea*, *Candidatus*, unclassified *Clostridiales*, and uncultured *Peptococcaceae* compared to the NP diet (*p* < 0.05; Table 6; Figure 1).

Real-time PCR quantification revealed that the total bacteria counts were not affected by the CP level in the colon digesta (*p* > 0.05). However, the counts of *Bacteroides*, *Clostridium Cluster IV*, and *Lactobacillus* showed a significant decrease (*p* < 0.05; Table 7) in the LP group compared to that in the NP group, and a moderate reduction of protein or extremely low-protein were both related to reduced *Ruminococcus* counts (*p* < 0.05; Table 7).

### 3.7. Microbial Metabolites in Colonic Lumen

In the colon, pigs fed the LP diet had significantly lower concentrations of acetate, isobutyrate, and isovalerate compared with those in the NP diet (*p* < 0.05; Table 8). Tryptamine and cadaverine were reduced in the MP and LP groups compared to the pigs in the NP group (*p* < 0.05), and an extremely LP diet decreased the concentration of putrescine (*p* < 0.05; Table 8). Additionally, significantly reduced concentrations of colonic ammonia were found in the LP group (*p* < 0.05; Table 8).

## 4. Discussion

Reducing dietary CP with free amino acid supplementation is a potential nutritional strategy for saving protein ingredients without impairing growth performance in pigs [22]. Conflicting results were found in the current researchers’ previous study: an extremely LP diet reduced growth performance, while moderate restriction of protein had no effect on weight gain and feed intake [5]. Thus, the response to an LP diet could be dependent on the degree of protein restriction. To investigate potential mechanisms underlying protein-dependent regulation of food intake and metabolism, blood parameters, digestive enzymes, intestinal epithelium morphology, intestinal endocrine function, and gut microbiota were tested in this study.

### 4.1. Glycemic Homeostasis

Dietary CP levels reduced by 6% of the NRC (1998), and supplementation with Lys, Met, Thr, and Trp resulted in reduced growth performance in pigs compared to pigs on the control diets [5]. Blood glucose levels remained normal when pigs were fed a low-protein diet for 16 weeks compared to the control group, indicating that glycemic homeostasis was well regulated. GIP is a well-known incretin hormones released from the gut in response to macronutrients and is responsible for amplified insulin release in a glucose-dependent manner [23]. In the present study, protein-restriction decreased plasma GIP levels, which was attributed to reduced feed intake. Ingested macronutrients needed to be digested by digestive enzymes to activate enteroendocrine cells that secrete GIP [24,25]. The current results showed that trypsin activity is not affected by an LP diet, suggesting that enzymatic function may not be the reason for reduced GIP in pigs. Notably, the activity of stomach pepsin and H^+^-K^+^-ATPase was increased by an LP diet, potentially promoting protein digestion. However, ingested proteins had little effect on incretin hormone secretion, while carbohydrates and fats potently facilitated gut hormone release [26,27]. The mechanism of protein sensing by enteroendocrine cells is poorly defined [23], and whether dietary protein level participates in blood glucose homeostasis requires further investigation.

### 4.2. Anorectic Effect

Extremely LP diets suppress food intake in humans and rodents [28], which was also shown in pigs in the researchers’ previous study [5]. Appetite is partly modulated by gut hormones secreted from the enteroendocrine system after sensing ingested feed. A large increment in PYY, an anorectic hormone, is associated with a high-protein diet compared to isocaloric high-carbohydrate or high-fat diets [29]. These results showed that reducing dietary protein does not decrease plasma PYY levels, despite significantly reduced feed intake induced by protein restriction. Additionally, PYY release can also be stimulated by fats [30,31]. Because dietary protein content was replaced with oil in LP groups, PYY was not reduced significantly by a protein-restricted diet in the present study. Reduced food intake and promoted satiety were also accompanied by an increase in plasma CCK levels with infusion of lipid emulsion into the ileum of humans [32,33], and CCK can also be stimulated by several amino acids and suppress feed intake in rats [34,35,36]. In the present study, although numerous amino acids and oil were included in the LP diet, the concentration of plasma CCK did not increase; thus, pigs in the LP group did not exhibit an obvious sense of hunger, which is also reflected by the decreased concentration of ghrelin in our results, an appetite-stimulating hormone.

### 4.3. Energy Metabolism

The circulating peptide leptin is mainly secreted by adipocytes and reflects the adipose tissue mass of the body [37]; adequate fat stores tend to secrete more leptin to increase energy expenditure; similarly, leptin levels fall after fasting and energy expenditure is reduced; thus, body-weight homeostasis can be controlled via leptin regulation [38]. In the study, plasma leptin levels were reduced in pigs fed an LP diet, suggesting that energy storage was not sufficient and that energy expenditure was likely diminished. In contrast, pigs fed LP diets provided fatter carcasses [39,40], partially due to increased carbohydrates and fats in LP diets [3]. These results showed that cholesterol levels were increased by a protein-restricted diet. Additionally, cholesterol synthesis can be induced by fatty acids [41] and turn into bile acids that facilitate lipid absorption in the gut lumen. However, a previous study showed that a low-protein diet tended to reduce back fat thickness (16.81 ± 2.73) compared to the control group (23.42 ± 2.17) [5]. These inconsistent results may be ascribed to impaired intestinal morphology and reducing energy intake under extremely LP diet conditions [7,42].

### 4.4. Gut Microbiota

Diet modulates the composition of gut microbiota, and long-term dietary habits have a considerable effect on gut microbiota [10]. The present study showed that *Streptococcus*, *Lactobacillus*, and *Bacteroides* are reduced, while *Prevotella* and *Ruminococcus* are increased in pigs, probably due to pigs’ diet being short of protein and composed predominately of corn and oil. Similar results showed that increased levels of *Bacteroides* and *Prevotella* are negatively correlated with energy intake and adiposity [43,44], suggesting that Bacteroidetes may be responsive to calorie intake and that gut microbiota have the potential to maximize energy absorption from a diet rich in carbohydrates. SCFAs are energy substrates for the colonic epithelium (butyrate) and peripheral tissues (acetate and propionate) [45], and the effect of the microbiota on energy deposition is dependent on the SCFA receptor [46]. However, the magnitude of energy produced by gut microbiota fermentation may not be enough to affect the host energy homeostasis, especially when *Lactobacillus* and SCFA are reduced in the colon, as shown in the present results. Thus, a shift in gut microbiota does not often occur [47,48,49], possibly due to varying experimental conditions, such as age, body weight, and duration of calorie restriction. Further meta-transcriptomic and proteomic studies should be conducted to provide insight into the response of microbial function as a result of dietary change. Apart from their use as energy substrates, SCFAs can also work as regulators, modulating secretion of the hormone GLP-1 by L-cells in the distal small intestine and colon [50], which improves insulin secretion effects. More evidence is needed to support the relationship between gut hormone and microbial metabolites in future research.

## 5. Conclusions

In summary, when presented with moderate dietary protein restriction, pigs are able to adjust their absorption and consumption of nutrients to maintain growth performance. However, extremely low-protein diets suppress appetite and reduce energy expenditure, although glucose homeostasis remains stable. Protein-restriction diets affect colonic microbial composition at the genus level, while bacteria diversity showed no significant difference. The production of microbial fermentation was decreased by extremely low-protein diets. However, the large number of pigs and the optimal length of different feeding periods should be considered when the effect of low-protein diets on the growth performance, gut development, and microbiota is investigated in future studies. Moreover, the production scale should also be included.

## Figures and Tables

**Figure 1 animals-09-00180-f001:**
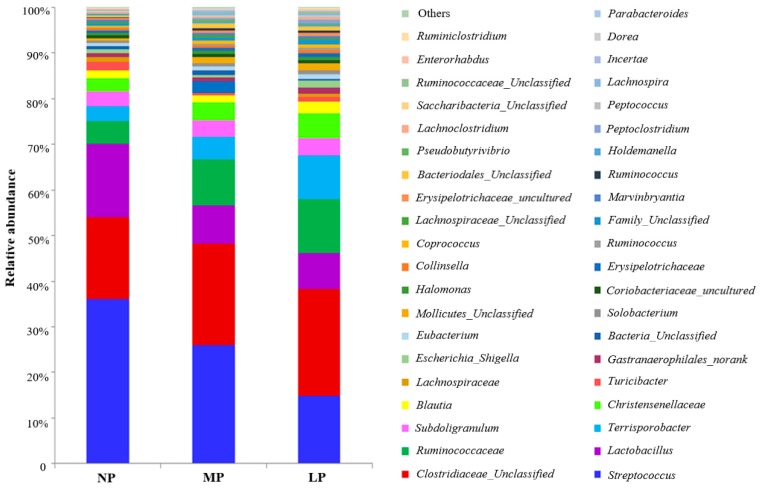
The relative abundance of microbial genera (with percentages greater than 0.1%) in the colon of pigs offered a normal protein (NP) diet, moderately low-protein (MP) diet, and a low-protein (LP) diet (*n* = 6).

**Table 1 animals-09-00180-t001:** Composition and nutrient level of experimental diets for pigs in different phases (as-fed basis).

Diets	Weaned Pigs			Growing Pigs			Finishing Pigs		
	NP	MP	LP	NP	MP	LP	NP	MP	LP
CP ^1^	20%	17%	14%	18%	15%	12%	16%	13%	10%
Corn	63.70	66.50	71.80	58.60	67.50	77.60	67.00	78.36	87.40
Soybean meal	19.80	18.80	13.40	29.00	19.50	10.00	23.76	15.00	5.50
Whey powder	4.30	4.30	4.40	-	-	-	-	-	-
Wheat bran	-	-	-	7.80	6.94	5.06	6.00	3.00	2.00
Fish meal	9.00	4.00	1.50	-	-	-	-	-	-
Soybean oil	0.80	2.60	4.10	1.55	2.38	3.00	0.88	0.90	1.71
Lysine	0.38	0.62	0.88	0.18	0.46	0.74	0.01	0.27	0.55
Methionine	0.10	0.19	0.27	0.00	0.09	0.17	0.00	0.00	0.09
Threonine	0.09	0.21	0.33	0.01	0.14	0.26	0.00	0.06	0.19
Tryptophan	0.01	0.04	0.08	0.00	0.02	0.07	0.00	0.01	0.06
CaHPO_3_	0.00	0.74	1.15	0.69	0.78	0.90	0.50	0.55	0.65
Rock powder	0.52	0.70	0.79	0.87	0.89	0.90	0.55	0.55	0.55
Salt	0.30	0.30	0.30	0.30	0.30	0.30	0.30	0.30	0.30
Premix ^2^	1.00	1.00	1.00	1.00	1.00	1.00	1.00	1.00	1.00
Total	100.00	100.00	100.00	100.00	100.00	100.00	100.00	100.00	100.00
DE (MJ/kg) ^3^	14.60	14.60	14.60	14.20	14.20	14.20	14.20	14.20	14.20
NE(MJ/kg) ^4^	10.49	10.72	10.98	10.26	10.54	10.80	10.39	10.64	10.91
Crude protein	20.05	17.09	14.09	18.27	15.16	12.35	16.30	13.17	10.26
Standardized ileal digestible (SID) AA									
Lysine	1.23	1.23	1.23	0.97	0.97	0.94	0.72	0.72	0.73
Methionine + Cystine	0.68	0.68	0.68	0.57	0.56	0.55	0.50	0.42	0.43
Threonine	0.73	0.73	0.73	0.61	0.61	0.60	0.56	0.50	0.49
Tryptophan	0.20	0.20	0.20	0.17	0.17	0.17	0.17	0.13	0.13
Arginine	1.09	0.90	0.68	1.09	0.83	0.58	0.95	0.71	0.46
Histidine	0.46	0.39	0.31	0.41	0.33	0.25	0.39	0.31	0.22
Isoleucine	0.70	0.59	0.45	0.64	0.49	0.35	0.60	0.45	0.30
Leucine	1.50	1.30	1.10	1.35	1.14	0.94	1.32	1.13	0.91
Phenylalanine	0.79	0.68	0.54	0.77	0.62	0.46	0.71	0.57	0.41
Valine	0.77	0.64	0.50	0.66	0.56	0.44	0.61	0.50	0.36
EAA ^5^	8.15	7.35	6.43	7.40	6.40	5.39	6.45	5.41	4.43
NEAA ^6^	8.72	7.47	5.98	8.97	7.26	5.49	8.02	6.33	4.61
EAA/NEAA	0.48	0.50	0.52	0.45	0.47	0.49	0.45	0.46	0.49

NP: normal protein diets according to the National Research Council (NRC, 2012); MP: moderately low-protein diet; LP: low-protein diet. ^1^ CP: crude protein. ^2^ For all of pigs, the premix provided these amounts of vitamins and minerals per kilogram on an as-fed basis: vitamin A, 10,800 international unit (IU); vitamin D3, 4000 IU; vitamin E, 40 IU; vitamin K3, 4 mg; vitamin B1, 6 mg; vitamin B2, 12 mg; vitamin B6, 6 mg; vitamin B12, 0.05 mg; biotin, 0.2 mg; folic acid, 2 mg; niacin, 50 mg; D-calcium pantothenate, 25 mg; Fe, 100 mg as ferrous sulfate; Cu, 150 mg as copper sulfate; Mn, 40 mg as manganese oxide; Zn, 100 mg as zinc oxide; I, 0.5 mg as potassium iodide; and Se, 0.3 mg as sodium selenite. ^3^ DE: digestible energy. ^4^ NE: net energy. ^5^ EAA: essential amino acid. ^6^ NEAA: non-essential amino acid.

**Table 2 animals-09-00180-t002:** Effect of dietary protein level on blood biochemical indices of pigs.

Item	NP	MP	LP	SEM	*p*-Value
Total protein, g/L	72.10	71.88	72.15	1.05	0.995
Albumin, g/L	39.42	39.63	42.62	1.10	0.438
Globulin, g/L	32.68	32.25	29.53	1.04	0.436
Blood urea nitrogen, mmol/L	5.02 ^b^	4.67 ^b^	2.63 ^a^	0.34	0.002
Glucose, mmol/L	5.57	5.33	5.52	0.26	0.936
Cholesterol, mmol/L	2.54 ^a^	2.52 ^a^	3.37 ^b^	0.16	0.043
Triglyceride, mmol/L	0.51	0.53	0.82	0.09	0.319

NP: normal protein diets according to the National Research Council (NRC, 2012); MP: reduced protein by 3% compared to the NP diet supplemented with Lys, Met, Thr, and Trp; LP: reduced protein by 6% compared to the NP diet supplemented with Lys, Met, Thr, and Trp. SEM: standard error of the mean. ^a,b^ Mean values in the same row differ in significance (*p* < 0.05).

**Table 3 animals-09-00180-t003:** Response of the digestive enzyme activity to different dietary protein levels in pigs.

Item	NP	MP	LP	SEM	*p*-Value
Pepsin, U/mL	20.90 ^a^	21.73 ^a^	31.98 ^b^	2.06	0.016
H^+^-K^+^-ATPase, U/mgprot	2.66 ^a^	2.73 ^a^	3.37 ^b^	0.13	0.029
Jejunal trypsin, U/mL	23560	28210	24097	2127	0.635
Ileal trypsin, U/mL	35082	32343	28107	1798	0.335

NP: normal protein diets according to the National Research Council (NRC, 2012); MP: reduced protein by 3% compared to the NP diet supplemented with Lys, Met, Thr, and Trp; LP: reduced protein by 6% compared to NP diet supplemented with Lys, Met, Thr, and Trp. SEM: standard error of the mean. ^a,b^ Mean values in the same row differ in significance (*p* < 0.05); mgprot: mg protein.

**Table 4 animals-09-00180-t004:** Effect of dietary protein level on small intestinal morphology of pigs, μm.

Item	NP	MP	LP	SEM	*p*-Value
Duodenum					
Villous height	616 ^a^	566 ^ab^	516 ^b^	15	0.011
Crypt depth	300 ^a^	268 ^ab^	254 ^b^	7	0.011
Villous: crypt	2.10	2.15	2.06	0.03	0.333
Jejunum					
Villous height	468	449	459	12	0.834
Crypt depth	200	215	232	6	0.082
Villous: crypt	2.38 ^a^	2.15 ^ab^	2.03 ^b^	0.05	0.013
Ileum					
Villous height	461	430	419	8	0.063
Crypt depth	235	216	197	7	0.091
Villous: crypt	2.01	2.05	2.21	0.06	0.342

NP: normal protein diets according to the National Research Council (NRC, 2012); MP: reduced protein by 3% compared to the NP diet supplemented with Lys, Met, Thr, and Trp; LP: reduced protein by 6% compared to the NP diet supplemented with Lys, Met, Thr, and Trp. SEM: standard error of the mean. ^a,b^ Mean values in the same row differ in significance (*p* < 0.05).

**Table 5 animals-09-00180-t005:** Effect of dietary protein level on gut hormones of pigs.

Item	NP	MP	LP	SEM	*p*-Value
GIP ^1^, ng/L	71 ^a^	59 ^b^	51 ^b^	2.65	<0.001
PYY ^2^, pg/mL	480	413	453	17.01	0.283
CCK ^3^, ng/L	123 ^a^	148 ^b^	113 ^a^	5.18	0.006
Ghrelin, ng/L	2439 ^a^	2273 ^b^	1882 ^b^	78.00	0.004
Leptin, μg/L	8.5 ^a^	7.5 ^a^	5.5 ^b^	0.34	<0.001
Gastrin, ng/L	108 ^a^	97 ^b^	76 ^b^	3.76	<0.001
Somatostatin, pg/mL	175 ^a^	159 ^b^	138 ^b^	5.08	0.003
5-HT ^4^, ng/mL	674 ^a^	531 ^b^	581 ^b^	19.97	0.004

NP: normal protein diets according to the National Research Council (NRC, 2012); MP: reduced protein by 3% compared to the NP diet supplemented with Lys, Met, Thr, and Trp; LP: reduced protein by 6% compared to NP diet supplemented with Lys, Met, Thr, and Trp. SEM: standard error of the mean. ^1^ GIP: glucose-dependent insulinotropic polypeptide. ^2^ PYY: peptide tyrosine tyrosine. ^3^ CCK: cholecystokinin. ^4^ 5-HT: serotonin. ^a,b^ Mean values in the same row differ in significance (*p* < 0.05).

**Table 6 animals-09-00180-t006:** The affected microbial genera in the colonic luminal contents of pigs fed different dietary protein levels, %.

Bacteria	NP	MP	LP	SEM	*p*-Value
*Streptococcus*	36.196 ^b^	25.544 ^ab^	13.405 ^a^	3.839	0.032
*Lactobacillus*	17.959 ^b^	8.241 ^a^	6.344 ^a^	2.154	0.042
*Turicibacter*	2.471 ^b^	0.503 ^a^	0.834 ^a^	0.076	0.039
*Prevotella*	0.069 ^a^	0.165 ^b^	0.141 ^b^	0.069	0.012
*Ruminococcus*	0.232 ^ab^	0.214 ^a^	0.383 ^b^	0.048	0.026
*Lachnospira*	0.067 ^a^	0.215 ^b^	0.584 ^c^	0.086	0.015
*Dorea*	0.109 ^a^	0.143 ^a^	0.391 ^b^	0.053	0.038
*Candidatus*	0.014 ^a^	0.027 ^a^	0.078 ^b^	0.084	0.013
Unclassified *Clostridiales*	0.006 ^a^	0.019 ^ab^	0.046 ^b^	0.035	0.031
Uncultured *Peptococcaceae*	0.004 ^a^	0.010 ^a^	0.024 ^b^	0.017	0.034

NP: normal protein diets according to the National Research Council (NRC, 2012); MP: reduced protein by 3% compared to the NP diet supplemented with Lys, Met, Thr, and Trp; LP: reduced protein by 6% compared to the NP diet supplemented with Lys, Met, Thr, and Trp. SEM: standard error of the mean. ^a,b,c^ Mean values in the same row differ in significance (*p* < 0.05).

**Table 7 animals-09-00180-t007:** Quantitative analysis of bacterial populations in the colonic digesta of pigs, Lg(copies/g wet content).

Bacteria	NP	MP	LP	SEM	*p*-Value
Total bacteria	9.87	9.94	9.68	0.076	0.396
*Bacteroides*	8.67 ^b^	8.27 ^ab^	8.07 ^a^	0.108	0.011
*Lactobacillus*	7.93 ^b^	7.64 ^ab^	7.41 ^a^	0.037	0.023
*Enterobacteriaceae*	7.42	7.33	7.31	0.105	0.907
*Clostridium Cluster XIV*	8.88	9.20	8.76	0.093	0.144
*Clostridium Cluster IV*	8.72 ^b^	8.51 ^ab^	8.22 ^a^	0.079	0.026
*Ruminococcus*	9.55 ^a^	10.05 ^b^	9.94 ^b^	0.077	0.015

NP: normal protein diets according to the National Research Council (NRC, 2012); MP: reduced protein by 3% compared to the NP diet supplemented with Lys, Met, Thr, and Trp; LP: reduced protein by 6% compared to the NP diet supplemented with Lys, Met, Thr, and Trp. SEM: standard error of the mean. ^a,b^ Mean values in the same row differ in significance (*p* < 0.05). Lg: log_10_(copies), Logarithmic Function.

**Table 8 animals-09-00180-t008:** Concentrations of short chain fatty acids, ammonia-N, and biogenic amines in the colonic digesta of pigs, µmol/g.

Item	NP	MP	LP	SEM	*p*-Value
SCFA					
Acetate	53.96 ^b^	45.84 ^ab^	36.70 ^a^	2.750	0.018
Propionate	23.64	20.49	19.67	1.110	0.497
Butyrate	12.28	11.93	11.26	0.820	0.882
Isobutyrate	1.73 ^b^	1.73 ^b^	0.71 ^a^	0.180	0.011
Valerate	3.29	3.61	3.13	0.310	0.841
Isovalerate	2.20 ^b^	2.10 ^b^	1.16 ^a^	0.200	0.011
Total SCFA	96.42 ^b^	85.75 ^ab^	73.46 ^a^	1.570	0.037
Ammonia-N	29.78 ^b^	26.37 ^ab^	16.63 ^a^	2.390	0.045
Biogenic amines					
Methylamine	0.31	0.20	0.29	0.039	0.515
Tryptamine	0.35 ^b^	0.17 ^a^	0.12 ^a^	0.033	0.010
Putrescine	1.52 ^b^	1.09 ^ab^	0.67 ^a^	0.130	0.016
Cadaverine	0.66 ^b^	0.27 ^a^	0.27 ^a^	0.076	0.044
Tyramine	0.21	0.15	0.21	0.041	0.805
Spermidine	0.40	0.27	0.17	0.054	0.237
Spermine	0.15	0.11	0.04	0.023	0.151

NP: normal protein diets according to the National Research Council (NRC, 2012); MP: reduced protein by 3% compared to the NP diet supplemented with Lys, Met, Thr, and Trp; LP: reduced protein by 6% compared to the NP diet supplemented with Lys, Met, Thr, and Trp. SEM: standard error of the mean. SCFA: short-chain fatty acids. ^a,b^ Mean values in the same row differ in significance (*p* < 0.05).

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
