# Peer review of "Effects of Long-Term Dietary Protein Restriction on Intestinal Morphology, Digestive Enzymes, Gut Hormones, and Colonic Microbiota in Pigs"

_animals, 2019, doi:10.3390/ani9040180_

Round 1
Reviewer 1 Report
This study is valuable for pig producers and nutritionists.
There are a few typos and grammatical errors, but I think they are a quick fix. Mostly the errors are in the abstract pertaining to grammatical tense.
In the abstract you could use the entire name for MP. Also, do you mean p>.05 or p<.05 on microbiota.
Line 43: essential amino acid vs indispensable aa.
Line 70: Shannon?
Results: You mention significant reduction in Streptococcus and Lactobacillus but only show the results and P value for Lactobacillus. I think it best if you include the Streptococcus. If it is p>.05, then include the values.
Author Response
Thank you very much for your comments about our manuscript submitted to Animals (Manuscript ID: animals-464427).
Those comments are all valuable and very helpful for revising and improving our manuscript, as well as the important guiding significance to our researches. We have studied comments carefully and made corrections which we hope meet with the approval. Revised portions are marked in yellow in the manuscript. We submit here the main corrections in the manuscript as well as a list of responds to the reviewer’s comments.
If you have any questions, please don’t hesitate to let us know.
Point 1: There are a few typos and grammatical errors, but I think they are a quick fix. Mostly the errors are in the abstract pertaining to grammatical tense.
Response 1: Thanks for the detailed comments. We are sorry for the mistakes in the abstract. The errors of typos and gramma have been revised in full paper, mainly in the abstract. The below is the revision of the abstract (the revised parts are marked in yellow):
Abstract: Using protein-restriction diets becomes a potential strategy to save the dietary protein resources. However the mechanism of low-protein diets influencing pigs’ growth performance is still controversial. This study aimed to investigate the effect of protein-restriction diets on gastrointestinal physiology and gut microbiota in pigs. Eighteen weaned piglets were randomly allocated to three groups with different dietary protein level. After a 16-week trial, the results showed that feeding a low-protein diet to pigs impaired epithelial morphology of duodenum and jejunum (p < 0.05), reduced the concentration of many plasma hormones (p < 0.05), such as ghrelin, somatostatin, glucose-dependent insulin-tropic polypeptide, leptin and gastrin. The relative abundance of Streptococcus and Lactobacillus in colon and microbiota metabolites was also decreased by extremely protein-restriction diets (p < 0.05). These findings suggested that long-term ingestion of a protein-restricted diet could impair intestinal morphology, suppress gut hormone secretion, and change the microbial community and fermentation metabolites in pigs, while the moderately low-protein diet had a minimal effect on gut function and did not impair growth performance. Please also see the details on Page 1, Line 22-35.
Point 2: In the abstract you could use the entire name for MP. Also, do you mean p>.05 or p<.05 on microbiota.
Response 2: Thanks for the reviewer’s comments. In the ABSTRACT, the “MP” has been changed into the entire name “moderately low-protein”, and the “p > 0.05” has been revised to “p < 0.05” after double checking. Please see the details in the revised manuscript (Page 1, Line 31 and 34).
Point 3: Line 43: essential amino acid vs indispensable aa.
Response 3: We agree with the reviewer’s comment. We have revised the “indispensable amino acids” to “essential amino acids”. Please see the details in the revised manuscript (Line 57).
Point 4: Line 70: Shannon?
Response 4: Thanks for the reviewer’s comments. Shannon is one of the indexes to describe the diversity of microbiota. In order to accurately describe it, the “bacterial Shannon diversity indices” was revised to “Shannon indices of bacterial diversity ”. Please see the details in the revised manuscript (Page 2, Line 76).
Point 5: You mention significant reduction in Streptococcus and Lactobacillus but only show the results and P value for Lactobacillus. I think it best if you include the Streptococcus. If it is p>.05, then include the values.
Response 5: Thanks for the reviewer’s detailed comments. After double checking, we found the P values for Lactobacillus and Streptococcus were all presented in the abstract and results. Please see the details in the following:
In the ABSTRACT, we showed that “the relative abundance of Streptococcus and Lactobacillus in colon and microbiota metabolites were also decreased by extremely protein-restriction diets (p < 0.05)”, P value stands for the significance of both Streptococcus and Lactobacillus.
In the RESULTS, the change of Streptococcus and Lactobacillus was described separately as following: At the genus level, the MP and LP diets significantly decreased the relative abundance of Lactobacillus and Turicibacter compared to those in the colon in the LP group (p < 0.05; Table 6; Figure 1). The relative abundance of Streptococcus was also reduced in the LP group compared to the NP group (p < 0.05; Table 6; Figure 1). Please also see details on Page 1, Line 30 and Page 7, Line 260-263.
Reviewer 2 Report
Introduction. If authors wanto to underline how important is the right feeding strategy to solve protein resources they should describe shortly different resolutions and recomended protein leveles in diets for growing pigs, taking into account additional amino acids supplementation.
It is important because: we recomend increasing protein level for fatteners. In europe we start from 16-18% for weaners and later we can use 3-4 diets with increasing concentration of crude protein. In this study and cited [5] there was used reverse way way - I don't understand why. I know that in tropic climate we start from higher protein level because animals have lower appetite.
The aim of this study is unclear for me. I understand that authors wanted to show how low protein level in daily ration influence on microbiota and pigs' growth performance - but what for? In this design I didn't found any useful advices.
Research design is appropiate - however for me it is totally unsuitable. It will be better to describe changes in animals' growth performance and microbiota when protein level is increasing in daily ration. Authors could look for the optimal lenght of different feeding periods and how it influences on investigated traits.
I have no comments to presented methods.
Results. Authors focus on microbiota and physiological parameters - ok, but there is no information about basis production parameters like daily gains, FCR - it should be included.
Conclusions. In presented study design they are correct and expected for me. Authors recomend future studies - but what they want to investigate?
Authors cited paper [5], where is described similar experimental design with lowering protein concentration in diets. It was concluded that such maintenance is unprofitable so why Authors performed such similar procedure - only to check microbiota and intestinal development?
Author Response
Thank you very much for your comments about our manuscript submitted to Animals (Manuscript ID: animals-464427).
Those comments are all valuable and very helpful for revising and improving our manuscript, as well as the significant guidance to our research. We have answered each of your points as below. Revised portions are marked in yellow in the manuscript.
If you have any questions, please don’t hesitate to let us know.
Point 1: To specify each of questions, we separate the questions in the first paragraph into Q1,Q2 and Q3.
Q1: Introduction: If authors want to underline how important is the right feeding strategy to solve protein resources they should describe shortly different resolutions and recommended protein levels in diets for growing pigs, taking into account additional amino acids supplementation.
Response Q1: Thanks for the reviewer’s valuable comments. Regarding to NRC 1998 recommendation, the requirement of crude protein is 20%, 18% and 16% for weaned piglets, growing pigs and finishing pigs, respectively. Previous researches showed that reducing the dietary protein level by less than 4% based on the NRC 1998, supplemented with Lys, Met, Thr, and Trp, did not reduced growth performance of weanling, growing, and finishing pigs (Figueroa et al. 2002; Htoo et al. 2007; Zhou et al. 2015). These findings indicate that a moderate reduction of dietary protein level (such as lower 4% than the NRC requirement) is an effective strategy to save the protein resource and decrease the emission of nitrogen in urea and faeces without impairing the growth performance in pigs. However, the latest edition of NRC 2012 recommends that the requirement of total nitrogen is 3.02%, 2.51%, 2.20%, 1.94% for piglets (11~25kg), small growing pigs (25~50kg), large growing pigs (50~75kg) and finishing pigs (50~75kg) respectively, which means that CP level is 2% ~ 4% lower than NRC 1998 calculated by the formula “CP = total nitrogen*6.25”. It is unclear whether the dietary protein level can be further reduced based on NRC 2012 and how low protein diets influence pigs’ growth performance, gut development and microbiota. Therefore, in the present study, to investigate the effect of different dietary protein level on pig growth, gut development and microbiota, different dietary protein level supplemented with four essential amino acids (Lys, Met, Thr, Trp) were applied based on NRC 2012.
Taking the reviewer’s suggestion, to highlight our research background and significance, in the revision of “INTRODUCTION” part, we made changes as the following: In 2016, China’s soybean imports were 8,391 million tons and accounted for more than 26% of the worldwide production, while high-protein (HP) diets led to excretion of excess nitrogen in faeces and urine, resulting in lowering the efficiency of nitrogen utilization and environmental pollution. NRC 1998 recommends that the requirement of crude protein is 20%, 18% and 16% for weaned piglets, growing pigs and finishing pigs respectively. Previous researches showed that reducing the dietary protein level by less than 4% based on the NRC 1998, supplemented with Lys, Met, Thr, and Trp, did not reduce growth performance of weanling, growing, and finishing pigs [2-5]. Thus, reducing dietary protein level is an effective strategy to save the protein resource and decrease the emission of nitrogen in urea and faeces without impairing the growth performance in pigs. However, the requirement of crude protein recommended by the latest edition of NRC 2012 is 2% ~ 4% lower than that of NRC 1998. Whether the dietary protein level can be further reduced based on NRC 2012 and the response of growth performance, gut development and microbiota to low-protein diets are unclear. Please also see the details on Page 1, Line 41-53.
Reference:
Figueroa JL, Lewis AJ, Miller PS, Fischer RL, Gómez RS, Diedrichsen RM. Nitrogen metabolism and growth performance of gilts fed standard corn-soybean meal diets or low-crude protein, amino acid-supplemented diets. J Anim Sci, 2002 80:2911–2919.
Htoo JK, Araiza BA, Sauer WC, Rademacher M, Zhang Y, Cervantes M, Zijlstra RT. Effect of dietary protein content on ileal amino acid digestibility, growth performance, and formation of microbial metabolites in ileal and cecal digesta of early-weaned pigs. J Anim Sci 2007, 85(12):3303–3312.
Zhou LP, Fang LD, Yue S, Yong S, Zhu WY. Effects of the dietary protein level on the microbial composition and metabolomic profile in the hindgut of the pig. Anaerobe, 2015, 38:61–69.
Q2: It is important because: we recommend increasing protein level for fatteners. In Europe we start from 16-18% for weaners and later we can use 3-4 diets with increasing concentration of crude protein. In this study and cited [5] there was used reverse way - I don't understand why. I know that in tropic climate we start from higher protein level because animals have lower appetite.
Response Q2: Thanks for the reviewer’s very valuable question. As the reviewer knows, we actually suggest our farmers that weaners should be fed 16-18% CP in China, and growing and finishing pigs were provided a lower protein in practice. Part of explanation is that both NRC 1998 and NRC 2012 recommendation of protein requirement are reduced linearly as pigs growing. The other explanation is that reducing the dietary CP by 2% to 4% supplemented with crystalline amino acids based on the NRC (1998) recommendation increased nitrogen utilization, reduced feed costs and nitrogen excretion, and promote gut health without impairing the growth performance (Figueroa et al. 2002; Htoo et al. 2007; Zhou et al. 2015), indicating that moderately reducing crude protein level is beneficial for pigs growing. In addition, for fatteners, feed mainly convert to body fat, which means a higher ingestion of protein cannot promote much muscle growth, indicating that feed is not used efficiently.
Q3: The aim of this study is unclear for me. I understand that authors wanted to show how low protein level in daily ration influence on microbiota and pigs' growth performance - but what for? In this design I didn't found any useful advices.
Response Q3: Thanks for the reviewer’s valuable comments. Our results showed that reducing the dietary protein by 3% based on NRC had no effect on growth performance, which means that more protein resource will be saved through using low-protein diets. However, the reasons for reduction of dietary protein without impaired the growth performance are unknown. Intestinal morphology and gut microbiota play an important role for nutrients digestion and absorption. Uncovering the changes of gut development and microbiota may provide the theoretical explanation for the growth performance in condition of low-protein diets. In terms of our results, reducing dietary protein level by 3% still maintains the gut development and microbiota diversity. These findings support that the homeostatic of gut development and microbiota contribute to maintain the growth performance under the condition of moderately reducing dietary protein (such as 3% lower than NRC).
As the reviewer’s suggestion, we made a clear “the aim of this study” in the revised manuscript, mainly in the part of “INTRODUCTION”. The following is the revision:
Gut homeostatic is important for pig growth. Thus, the aim of this study was to investigate effects of long-term dietary protein restriction on intestinal morphology, digestive enzymes, gut hormones and colonic microbiota in pigs, which may provide a theoretical explanation for the change of growth performance under the condition of reducing dietary protein. This study is expected to provide a good foundation for future diet formula in practical use and contributes to save a large quantity of protein. Please also see the details on Page 2, Line 83-88.
Point 2: Research design is appropriate - however for me it is totally unsuitable. It will be better to describe changes in animals' growth performance and microbiota when protein level is increasing in daily ration. Authors could look for the optimal length of different feeding periods and how it influences on investigated traits.
Response 2: Thanks for reviewer’s very good suggestions. Actually, our experiment were conducted by us and our partner Tiejun Li (Institute of Subtropical Agriculture, The Chinese Academy of Science). Tiejun Li is responsible for feed formula and the evaluation of pigs’ growth performance under the condition of low-protein diets, while we mainly investigate the effect of low-protein diets on gut development and microbiota. Therefore, the data patent of growth performance belongs to Tiejun Li (unpublished data), we briefly describe it in our manuscript, please see the details in Line 208 of the revised manuscript.
Analysis of gut microbiota for finishing pigs(from weaning to fishing lasting for 16 weeks trial)was shown in Table 6, Table 7 and Figure 1 in our manuscript. Briefly, low-protein diets decreased the relative abundance of Streptococcus, Lactobacillus and Turicibacter in the colon, while increased the relative abundance of Prevotella, Lachnospira, Dorea, Candidatus, unclassified Clostridiales and uncultured Peptococcaceae.
However, for weaned piglets(lasting for 5 weeks after weaned), the data of microbiota was submitted to another journal (Archives of Animal Nutrition) and the manuscript was under reviewed currently. The following is the main results: At the phyla level, Firmicutes are predominant in jejunum and colon. At the genus level, Weissella and Lactobacillus were predominant in jejunum, while Clostridium_sensu_stricto_1, lactobacillus, and peptostreptococcaceae_unclassified were predominant in colon. In phyla and genera, jejunum and colon microbiota was not affected by low protein or moderately low-protein diets.
Combined with our results, the change of gut microbiota and growth performance depend on the length of feeding periods. However, the optimal time point of the change in microbiota and its association with grow performance is not well revealed. We agree with the reviewer’s suggestion, to find the optimal length of different feeding periods, digesta samples will be collected at more time points for analysis of dynamic changes in gut microbiota and its association with growth performance in our further study.
Remark:
The data patent of ADG, FI and FC in this experiment belongs to our partner Tiejun Li and the data has not been published yet. Therefore, we briefly describe it instead of a detailed Table. We are very sorry for this.
Point 3: Results: Authors focus on microbiota and physiological parameters - ok, but there is no information about basis production parameters like daily gains, FCR - it should be included.
Response 3: Thanks for the reviewer’s suggestion. We have added “the growth performance” in the RESULTS. The following is the revision: From week 5 to 16, low-protein (LP) diets significantly reduced body weight, average daily gain and feed intake in piglets, growing, and finishing pigs compared with normal protein (NP) diets. The ratio of feed intake to weight gain was decreased in piglets, growing pigs in low-protein (LP). However, moderately low-protein (MP) diets had no effect on body weight, average daily gain and feed intake during the whole period in pigs (unpublished data). Please also see the details in the revision (Line 211).
Here, we would like to make a declaration that the data patent of ADG, FI and FC in this experiment belongs to our partner in the program (the National Key Basic Research Program of China, grant number 2013CB127301). Therefore, a brief description of the growth performance is introduced in our revised manuscript instead of a clear Table. Please see the details in Line 208.
Point 4: Conclusions: In presented study design they are correct and expected for me. Authors recommend future studies - but what they want to investigate?
Response 4: Thanks for the reviewer’s good comments. We are sorry for the unclear recommendations for the future studies. Actually, in our present study, we found that gut development and microbiota were not impaired when pigs were fed moderately low-protein diets (3% lower that NRC 2012) for a long-term period, while the growth performance of pigs can also be maintained. However, for practical use, the number of pigs in each group (6 pigs) involved is still small, and the results should be reproducible. Therefore, a large number of pigs should be included for the future study. Moreover, as reviewer suggest before, the optimal length of different feeding periods should be considered when pigs are fed a long-term low-protein diet. Finally, in different farms with different sanitary condition and different production scale, the effect of low-protein diets on the gut morphology and microbial composition needs to be investigated as well.
Therefore, we have added “details of further investigation” in the part of “CONCLUSIONS”. The revised parts have been marked in yellow, the following is the revision:
In summary, when presented with moderate dietary protein restriction, pigs are able to adjust their absorption and consumption of nutrients to maintain growth performance. However, extremely LP diets suppress appetite and reduce energy expenditure, although glucose homeostasis remains stable. Protein-restriction diets affect colonic microbial composition at the genus level, while bacteria diversity showed no significant difference. The production of microbial fermentation was decreased by extremely LP diets. However, the large number of pigs and the optimal length of different feeding periods should be considered when the effect of low protein diets on the growth performance, gut development and microbiota is investigated in the future studies. Moreover, the production scale should also be included. Please also see the details on Page 11, Line 381-384.
Point 5: Authors cited paper [5], where is described similar experimental design with lowering protein concentration in diets. It was concluded that such maintenance is unprofitable so why Authors performed such similar procedure - only to check microbiota and intestinal development?
Response 5: Thanks for the reviewer’s valuable comments. We are sorry for the unclear description and objectives. Actually, the diet formula of cited paper [5] was different from ours. Theirs was formulated based on NRC 1998. Ours was formulated based on NRC 2012. Their results showed that reducing dietary protein by 3% had no effect on growth performance. Currently, NCR 2012 is available. Whether the dietary protein can be reduced based on NRC 2012 without impaired growth performance and gut health is unknown.
The aim of our present study is to investigate the effects of reducing dietary protein level based on NRC 2012 on the gut development and microbiota, which may provide a good foundation for future diet formula in practical use and save a large quantity of protein.
Round 2
Reviewer 2 Report
After corrections I like this manuscript.
-line 213 in brackets: shouldn't be: "confidential data" ?
-line 352 shouldn't be "a low protein diet"?
I have no more comments. Good luck
Author Response
Dear reviewer,
Thank you for your comments and suggestions. We provide a point-by-point response to the reviewer’s comments, please see the details below.
Point 1: line 213 in brackets: shouldn't be: "confidential data" ?
Response 1: Thanks for the reviewer’s valuable comments. The data patent of growth performance belongs to our partner Tiejun Li, who have not yet published it in detail. So the data is still confidential. Taking the reviewer’s suggestion, we revise the “unpublished data” to “confidential data” in brackets, please see the details in Page 6, Line 213.
Point 2: line 352 shouldn't be "a low protein diet"?
Response 2: Thanks for the reviewer’s valuable comments. Exactly, “An low protein diet” is a grammatical mistake. As the reviewer’s suggestion, we revise the “An low protein diet” to “a low protein diet”. Please see the details in Page 10, Line 352.